# Lower Limb Kinematic Coordination during the Running Motion of Stroke Patient: A Single Case Study

**DOI:** 10.3390/jfmk7010006

**Published:** 2022-01-06

**Authors:** Noboru Chiba, Tadayoshi Minamisawa

**Affiliations:** 1Department of Occupational Therapy, Yamagata Prefectural University of Health Sciences, 260 Kamiyanagi, Yamagata 990-2212, Japan; 2Department of Physical Therapy, Yamagata Prefectural University of Health Sciences, 260 Kamiyanagi, Yamagata 990-2212, Japan; tminamisawa@yachts.ac.jp

**Keywords:** running, coherence analysis, stroke patient, kinematic, lower limb joint coordination

## Abstract

The purpose of this study was to clarify the lower limb joint motor coordination of para-athletes during running motion from frequency characteristics and to propose this as a method for evaluating their performance. The subject used was a 43-year-old male para-athlete who had suffered a left cerebral infarction. Using a three-dimensional motion analysis system, the angles of the hip, knee, and ankle joints were measured during 1 min of running at a speed of 8 km/h on a treadmill. Nine inter- and intra-limb joint angle pairs were analyzed by coherence and phase analyses. The main characteristic of the stroke patient was that there were joint pairs with absent or increased coherence peaks in the high-frequency band above 4 Hz that were not found in healthy subjects. Interestingly, these features were also observed on the non-paralyzed side. Furthermore, a phase analysis showed different phase differences between the joint motions of the stroke patient and healthy subjects in some joint pairs. Thus, we concluded there was a widespread functional impairment of joint motion in the stroke patient that has not been revealed by conventional methods. The coherence analysis of joint motion may be useful for identifying joint motion problems in para-athletes.

## 1. Introduction

The quantitative and objective evaluation of human running is considered to have many advantages in terms of improving performance and offering a basic understanding of the complex mechanism involved in running [1]. Therefore, appropriate movement analysis is a very effective means for improving the performance of athletes. Furthermore, the significance of movement analysis is even greater for para-athletes who are stroke patients, as motor palsy, a central nervous system disorder, results in the loss of a wide range of voluntary motor skills in the body [2]. In fact, it has been suggested that elucidating individual muscle activity patterns [3] and joint stiffness [4] in complex movements such as running may lead to improvements in running behavior. Furthermore, it may provide useful information for the prevention of tendon disorders [5] that may have a negative impact on the performance of athletes and for rehabilitation approaches to muscle injuries [6]. In the analysis of the running movements of competitive athletes, the collection of joint moments [7,8], joint angles [9], and ground reaction forces [10] through motion capture systems as well as the collection of lower limb muscle activity through electromyography [11] mechanics work [12] and mechanical work efficiency [13] are commonly carried out in analyses. In the first place, coordination problems in addition to ROM and muscle weakness have been pointed out as factors contributing to poor performance in stroke patients, but to the best of our knowledge no study has clarified the motor coordination of these lower limb joints during running. Human motor performance is ultimately transmitted from the motor cortex to the muscle groups through the spinal cord circuit to output the movement [14]. Therefore, knowing the effect of multi-joint motion on coordination in whole-body forms of exercise such as running may lead to gaining an essential understanding of performance. Therefore, it is necessary to analyze the interaction of the entire lower limb rather than just focusing on a localized area [15]. Previous studies have shown how to construct a functional muscle network by assessing inter-muscular coherence from surface electromyograms (EMG) recorded from different muscles [16]. Such a functional network seems to be able to reveal functional connections between groups of muscle cords in multiple frequency bands. This is because coherence between EMGs shows correlations or common inputs to spinal motoneurons, which are generated by covalent structural coupling or synchronization within the motor system [17,18]. We propose to extend this method to reveal the interaction between the two lower limbs as the sum of motor outputs under the control of the cerebral cortex by inter- and intralimb coherence analysis in joint movements, which is so far unknown. Normally, the left and right lower limbs or joints within must always be coordinated in order to maintain stability in activities such as walking [19] and running [20]. Although the overall picture of inter-limb coordination in running motion remains unclear, the background is probably based on a neuronal network that generates the rhythmic activation of muscles controlling the limbs [21], which is also the neural basis of running. For tighter coupling in left and right leg movements, such interneurons in the spinal cord allow direct communication to occur between the left and right sides without the need for the involvement of higher centers. On the other hand, given that lower limb muscle activity during walking is strongly controlled by the cortex [22,23], the same involvement by the primary motor cortex is likely to occur in running movements. Therefore, in stroke patients, such cortical control of muscle activity may not be sufficiently performed, which may inhibit running movements. The viewpoint of inter-limb and intra-limb motor coordination given in the present study may provide essential and important information about running motion that has been lacking in previous studies.

The purpose of this study was to examine problems in the running motion of stroke patients by clarifying the coherence characteristics of joint angles during running in para-athletes and to propose an analysis method that can be used to evaluate the performance of athletes in running.

## 2. Materials and Methods

### 2.1. Participant

The participant of this experiment was a male with left cerebral hemorrhage (age: 43 years; height: 167 cm; body mass: 57 kg) who experienced a left stroke 6 years ago and became right hemiplegic. He was an athlete who also competed in para-triathlons. Fugel-Mayer Assessment (FMA) scores were 56/66 for upper limb, 25/34 for lower limb, 14/14 for balance, 19/24 for sensation, 40/44 for joint range of motion, and 42/44 for joint pain. Additionally, Brunnstrom Recovery Stage Test showed that he had a Stage V impairment in his upper and lower extremity function, but he was independent in his daily life and drove a car daily. In this study, one healthy young adult male (age: 41 years; height: 178 cm; body mass: 67 kg) was also included as a comparison subject.

### 2.2. Measurement Method

In this study, changes in the angles of the hip, knee, and ankle joints were collected continuously for one minute while the subjects were running on a treadmill. The walking speed was determined to be 8 km/h, which is a sprint speed, since the two subjects in this study were athletes who trained daily. The measurements of joint angles were carried out using a three-dimensional motion analysis system with 6 cameras (MX T-20, Vicon Motion Systems Ltd., Oxford, UK). To calculate the joint angles, 35 infrared reflective markers were attached to the subjects’ bodies and a plug-in gait full-body model was used. The measurement data were subjected to a 12 Hz low-pass filter using a fourth-order Butterworth filter. All data analyses were performed using the NI DIAdem 2020 (National Instrument, Austin, TX, USA).

### 2.3. Analysis of Joint Movement Coordination

The degree to which the modulation of two joint angle pairs in the sagittal plane during treadmill running was correlated was analyzed in the frequency range of 0–12 Hz using the coherence method. As inter-limb pairs, we analyzed the coherence at the right and left hip joints, the right and left knee joints, and the right and left ankle joints. As intra-limb joint pairs, the right hip and right ankle, right hip and right ankle, right knee and right ankle, and joint pairs within the left lower limb were analyzed as well.

The coherence between two signals is a normalized measure of the linear correlation between signals in the frequency domain, with values ranging from 0 to 1, where 1 indicates perfect linear correlation and 0 indicates no linear correlation at all [17,24]. The joint angle coherence was estimated between pairs of concatenated joint angle data using the Welch method with a window length of 333 points and 0% overlap.
$$\left|C_{xy}\left(f\right)\right|=\frac{{\left|S_{xy}\left(f\right)\right|}^{2}}{S_{xx}\left(f\right)S_{yy}\left(f\right)}$$
where *Sxy* (*f*) and *Sxx* (*f*) represent the CPSD and *Sxx* (*f*) and *Syy* (*f*) represent the PSD of both input signals *x(t)* and *y(t)*, respectively—i.e., any pairwise combination of the joint angles under investigation. Moreover, coherence is considered to be significant if the resulting value lies above the confidence level (CL) [25]. In this study, the threshold coherence value was set to 0.31.

In this study, the phase values were limited to −180° to +180°. A positive phase indicates that the target signal was leading the reference signal. For the inter-limb joint angles, the reference signal was the joint angle of each right lower limb, and the target signal was the joint angle of the left lower limb. Intra-limb joint angles were analyzed using the more proximal joint (e.g., hip joint in the case of the hip–knee joint) as the reference signal.

## 3. Results

Figure 1 shows the joint angles of the lower extremities (hip, knee, and ankle) during running in the stroke patient and the healthy subject.

### 3.1. Interlimb Coherence and Phase Analysis

In the case of the stroke patient, the coherence peak was found to be partially absent in the frequency band above 4 Hz in the hip inter-limb pairs a and a’ or b and b’ in Figure 2. On the other hand, in the stroke patient peaks were observed in the frequency band that were not present in healthy subject c and c’ in Figure 2. The phase characteristics of the limb movements were similar in the stroke patient and healthy subject, but some phases were different in the two groups (see d,d’ in Figure 2).

### 3.2. Intra-Limb Coherence Analysis and Phase Analysis (Left Limbs)

For the left lower limb (i.e., non-paralyzed side) intra-limb coherence of the stroke patient, the overall coherence peak was found to be similar to that of healthy subject. On the other hand, the hip–knee pair and the hip–ankle pair showed coherence peaks not seen in normal subject (a,a’,b,b’ in Figure 3). In terms of phase, the phase of the stroke patient was almost zero in the hip–knee pair (c,c’ in Figure 3).

### 3.3. Intra-Limb Coherence Analysis and Phase Analysis (Right Limbs)

The intra-limb coherence of the right lower limb (non-paralyzed side) of the stroke patient showed a deficit of coherence peaks at frequencies higher than 4 Hz compared to the healthy subject (a,b,c in Figure 4A, a’,c’,d’ in Figure 4B). On the other hand, a coherence not seen in healthy subject was observed, with a broad peak occurring at 10–12 Hz (b,b’ in Figure 4).

As for the phase, the results for e and e’ of the hip–knee joint were similar to those for the left limb, and in the case of stroke patient, the phase was almost zero in the high-frequency band.

## 4. Discussion

We found that the coherence of inter- and intra-extremity joint angles in the running motion of the stroke patient had two characteristics, with the first being the partial absence of coherence peaks in the frequency bands above 4 Hz. This was one of the characteristics of the stroke patient, since the healthy subject showed clear inter-limb coherence peaks in a relatively large number of frequency bands (e.g., a in Figure 2). A possible reason for this is the reduced muscle activity in the paralyzed lower limbs, which is characteristic of stroke patients [26,27]. When the plantar flexors do not function properly during running, they are unable to generate propulsive force, which is an important component of athletic performance [28], meaning that inter- and intra-extremity coordination, as seen in healthy individuals, is impaired. Given that running is an inter-joint coordination exercise, the partial decrease in coherence values observed in the stroke patient may represent a decrease in the limb coupling ability.

Secondly, it is also interesting to note that the peak of coherence was observed in the stroke patient and was not seen in the healthy subject. It has been shown that in rhythmic movements, such as human walking, the frequencies of body acceleration are usually observed in the lower range [29]. Considering that higher frequencies are observed in pathological subject, it is also interesting to note that a peak in coherence was observed in stroke patient, but not in healthy subject. Specifically, as shown in a of Figure 3 and b of Figure 4, the intra-limb coherence of the hip and knee joints at around 10 Hz showed high peaks in both the left and right lower limbs, which is one of the features not seen in healthy subject. As a background, it is possible that in the present case, the proximal joints are actively involved in the stride to compensate for the reduced coherence of the distal paralyzed limb. Since running movements in stroke patients require the use of more hip flexor muscle groups to maintain speed [30], the increase in peak coherence could be the result of the active movement of the hip joint relative to the adjacent knee joint. A possible neural mechanism for this is co-activation. The phenomenon of co-activation has been explained by its effects on kinematics (e.g., an increase in the apparent stiffness of the joint, the facilitation of faster movements, effects on behavioral stability), its impact on movement optimization, and the involvement of various neurophysiological structures [3]. The requirement for quick movements during running may reflect the neural processes that adapt to such movements [31]. On the other hand, the excessive co-activation of muscle groups involved in joint movements may inhibit abnormal patterns of muscle activity and redundancy of movement [32], and such enhanced coherence may not be a desirable state. In other words, as a background for the increased coherence of the proximal joints of both lower limbs, this may be a strategy to compensate for the motor impairment of the distal part by the proximal joints of the paralyzed limb and to maintain a symmetrical posture by the lower limbs of the non-paralyzed limb [33]. Previous reports on mechanical energy during walking in stroke victims have reported the presence of neural connections between limbs, such as increased movement on the non-paralyzed side to stabilize the body against asymmetries in internal energy generation after stroke. Such a concept is probably also possible in running. Therefore, the results of this study in running may reflect the complex neural adaptations of the motor system seen after stroke [34].

To begin with, in human walking locomotion, CPGs coordinate and control the many skeletal muscles that participate in the movement. Although the neural basis of running locomotion has not been widely discussed, its control mechanism is not considered to be significantly different from that of locomotion control during walking, in which spinal nerves are involved [35]. On the other hand, considering the fact that stroke patients have gait disorders, it is clear that the cerebral cortex contributes to locomotion. Therefore, the assessment of joint coordination by coherence analysis is able to reliably discriminate stroke patient from young adults. In other words, such a method should likely be further explored as a performance evaluation for para-athletes who are stroke patients.

In this study, we also conducted a phase analysis and confirmed that there is a phase difference in each joint pair of lower limb movements in stroke patients. In particular, the phase differences represented by the paired parts, in Figure 2 such as a and a’ suggests that the stroke patient was performing a different motor strategy from the healthy person. Although it was unclear to what extent such differences affected the performance of both subjects, such features may affect the interaction of joint movements necessary for forward propulsion.

A limitation of this study is that the relationship between the coherence of joint motion and each phase of the running cycle (stride, swing, etc.) is unknown. Determining which phases are affected by the peak coherence problems in brain-injured individuals shown in this study will bring us even closer to solving the problems associated with running in para-athletes.

In this study, a motion capture system was used to evaluate the motion coordination. In fact, the 3D motion analysis system used in this experiment was very expensive, but recently, a very inexpensive gait analysis tool that evaluates joint angles has been proposed [36]. Therefore, evaluation outside the laboratory can be expected. Furthermore, we believe that our proposed method for analyzing the coordination of joint motion can be widely applied not only to the running evaluation of para-athletes and mild stroke patients, but also to normal gait analysis.

Stroke patients suffer from a variety of neurological symptoms and secondary motor disorders [37]. Therefore, in order to improve their performance, it is necessary to analyze their individual performance, and this kind of analysis method may contribute to the improvement of para-athletes’ competitive performance. On the other hand, the neural mechanisms of the central nervous system, including the cortex of inter-limb coordination in running behavior, have not yet been elucidated. Therefore, this kind of research will be important in the future.

## Figures and Tables

**Figure 1 jfmk-07-00006-f001:**
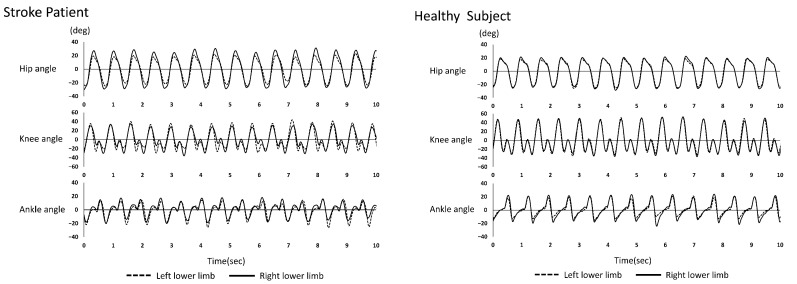
Representative examples of joint angles of the lower limbs during running in the stroke patient and the healthy subject are shown. The **left** figure shows the stroke patient, and the **right** figure shows the healthy subject. The upper panel shows the hip joint, the middle panel shows the knee joint, and the lower panel shows the ankle joint. Note that the left and right joint angles were shown superimposed after the time axis was matched.

**Figure 2 jfmk-07-00006-f002:**
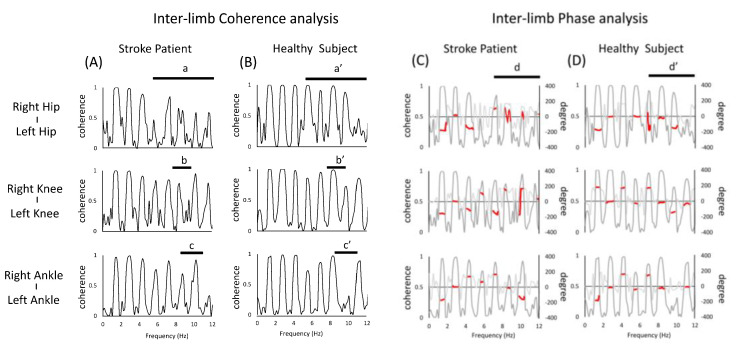
Diagram of coherence analysis and phase analysis between inter-limbs. The figure (**A**,**B**) shows the results of the coherence analysis. The figure (**C**,**D**) shows the results of the phase analysis overlaid with the results of the coherence analysis (gray line). When the peak coherence is greater than 0.3, the phase results are partially shown with red lines. The top, middle, and bottom rows show the hip, knee, and ankle joint pairs, respectively. In the figure, (a,a’), (b,b’), (c,c’) shows coherence analysis, (d,d’) shows the different parts of the phase analysis between stroke patient and healthy subject.

**Figure 3 jfmk-07-00006-f003:**
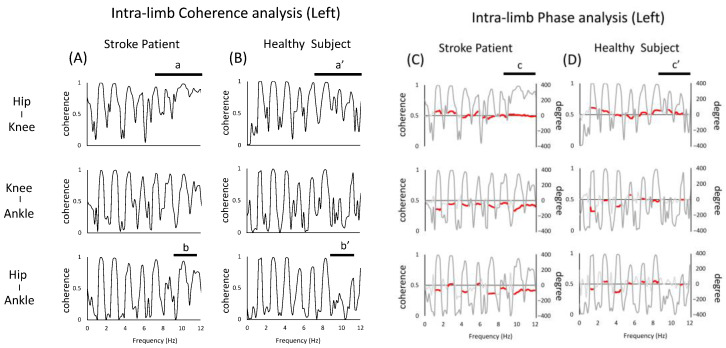
Diagram of the intra-limb (left limb) coherence and phase analysis of the left lower limb. The figure (**A**,**B**) shows the results of the coherence analysis. The figure (**C**,**D**) shows the results of the phase analysis overlaid with the results of the coherence analysis (gray line). When the peak coherence is greater than 0.3, the phase results are partially shown with red lines. The top, middle, and bottom rows show the hip, knee, and ankle joint pairs, respectively. In the figure, (a,a’), (b,b’) shows coherence analysis, (c,c’) shows the different parts of the phase analysis between stroke patient and healthy subject.

**Figure 4 jfmk-07-00006-f004:**
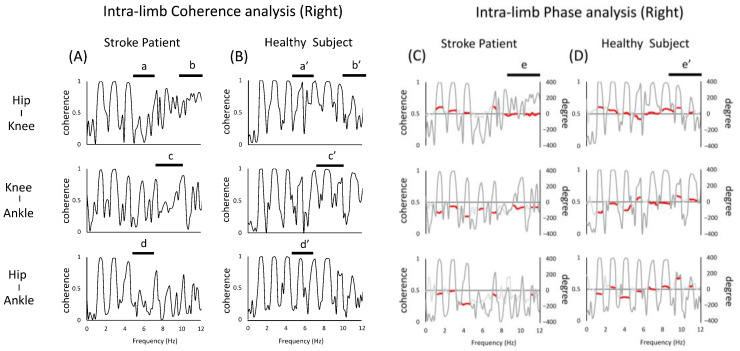
Diagram of the intra-limb (right limb) coherence and phase analysis of the left lower limb. The figure (**A**,**B**) shows the results of the coherence analysis. The figure (**C**,**D**) shows the results of the phase analysis overlaid with the results of the coherence analysis (gray line). When the peak coherence is greater than 0.3, the phase results are partially shown with red lines. The top, middle, and bottom rows show the joints representing the hip, knee, and ankle joint pairs, respectively. In the figure, (a,a’), (b,b’), (c,c’), (d,d’) shows coherence analysis, (e,e’) shows the different parts of the phase analysis between stroke patient and healthy subject.

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
