# Peer review of "Lower Limb Kinematic Coordination during the Running Motion of Stroke Patient: A Single Case Study"

_jfmk, 2022, doi:10.3390/jfmk7010006_

Round 1

Reviewer 1 Report

I think, the overall goal of rehabilitation in Stroke subjects is walking with no limitation. after this, the running is important, therefore the running may be is not interesting in rehab readers and stroke subjects.

please explain more about inclusion and exclusion criteria in the method section.

in addition please clarify how sample size was detected ?

please more about clinical finding and usage of results in clinical rehabilitation in the discussion section.

Author Response

We are thankful for the essential questions and suggestions of the reviewers regarding the research contents. Our responses to the comments are as follows:

Comments:

I think, the overall goal of rehabilitation in Stroke subjects is walking with no limitation. after this, the running is important, therefore the running may be is not interesting in rehab readers and stroke subjects.

Response:

We think you are right in pointing out that the hope of many stroke patients is to achieve safe walking. On the other hand, sporting opportunities for individual with physical disabilities have increased dramatically in the last few decades [Refer]. We believe that there is a certain amount of people with relatively mild strokes who need to compete at a higher level as para-athletes or who run as a hobby. In other words, if a person has a physical problem due to stroke, it is necessary to analyze and teach how to prevent secondary disabilities through sports at any level. Therefore, we believe that our report can contribute to the development of longer-term sports careers and leisure activities for such individual.

Furthermore, in addition to the fact that athletes participate in the Paralympic Games for a sense of accomplishment and to enhance their abilities, the authors also state that athletes have made close friends as a result of their participation and that demonstrating their abilities to others is a strong motivation for their participation. In this respect, we believe that it can contribute to the improvement of QOL.

[Refer] Jefferies P, Gallagher P, Dunne S. The Paralympic athlete: a systematic review of the psychosocial literature. Prosthet Orthot Int. 2012 Sep;36(3):278-89. doi: 10.1177/0309364612450184. PMID: 22918904.

Comments:

please explain more about inclusion and exclusion criteria in the method section.

Response:

Regarding the exclusion criteria you mentioned, this study is a single-case design, so there are no clear exclusion criteria, but we made sure that the patients did not have any musculoskeletal disorders or progressive neurological symptoms before they entered the study.

Comments:

in addition please clarify how sample size was detected ?

Response:

We apologize for the misleading title and the inadequate explanation of our methodology. This report is a single case study. Therefore, there will be one stroke patient and one control person. The title has been revised to clearly state that it is a single-case design in the methods section.

Comments:

please more about clinical finding and usage of results in clinical rehabilitation in the discussion section.

Response:

We have added the following: (Page 6, Lines 229-233)

“In fact, the 3D motion analysis system used in this experiment was very expensive, but recently, a very inexpensive gait analysis tool that evaluates joint angles has been proposed [34]. Therefore, evaluation outside the laboratory can be expected. Furthermore, we believe that our proposed method for analyzing the coordination of joint motion can be widely applied not only to the running evaluation of para-athletes and mild stroke patients, but also to normal gait analysis.”

[34] Bahadori S, Davenport P, Immins T, Wainwright TW. Validation of joint angle measurements: comparison of a novel low-cost marker-less system with an industry standard marker-based system. J Med Eng Technol. 2019 Jan;43(1):19-24. doi: 10.1080/03091902.2019.1599072. Epub 2019 Apr 29. PMID: 31033375.

Reviewer 2 Report

This is a very interesting article about Lower Limb Kinematic analysis in a patient suffered from stroke.

I want to congratulate you for your work, since it is well written, clear and give an interesting insight into para-athletes characteristics

I suggest to see CARE checklist (https://www.care-statement.org/checklist)  and try to improve your work following their points: title, keywords, abstract, introduction, patient information, clinical findings, timeline, diagnostic assessment, therapeutic intervention, follow-up and outcomes, discussion, patient perspective, informed consent

Please consider the following references:
- https://www.ncbi.nlm.nih.gov/pmc/articles/PMC7433331/ --> walking gait analysis has already been studied and this is an interesting aspect to apply
- http://rua.ua.es/dspace/handle/10045/108922 --> your lower limb kinematic analysis is crucial also to avoid tendinopathies (http://rua.ua.es/dspace/handle/10045/108922) and muscle injuries in such athletes (https://www.mdpi.com/2411-5142/6/3/75) 

Author Response

We are thankful for the essential questions and suggestions of the reviewers regarding the research contents. Our responses to the comments are as follows:

Comment:

I suggest to see CARE checklist (https://www.care-statement.org/checklist)  and try to improve your work following their points: title, keywords, abstract, introduction, patient information, clinical findings, timeline, diagnostic assessment, therapeutic intervention, follow-up and outcomes, discussion, patient perspective, informed consent

Response:

Thank you for your suggestion. We have added "patient information", "diagnostic assessment", which were inadequate in our previous manuscript.

We have added the following: (Page 2, Lines 80-88)

The participant of this experiment was a male with Left cerebral hemorrhage  (age: 43 years; height: 167 cm; body mass: 57 kg) who experienced a left stroke 6 years ago and became right hemiplegic. He was an athlete who also competed in para-triathlons. Fugel-Mayer Assessment (FMA) scores were 56/66 for upper limb, 25/34 for lower limb, 14/14 for balance, 19/24 for sensation, 40/44 for joint range of motion, and 42/44 for joint pain. Also, Brunnstrom Recovery Stage Test showed that he had a Stage V impairment in his upper and lower extremity function, but he was independent in his daily life and drove a car daily. In this study, one healthy young adult male (age: 41 years; height: 178 cm; body mass: 67 kg) was also included as a comparison subject.

Comment:

Please consider the following references:

- https://www.ncbi.nlm.nih.gov/pmc/articles/PMC7433331/ --> walking gait analysis has already been studied and this is an interesting aspect to apply

- http://rua.ua.es/dspace/handle/10045/108922 --> your lower limb kinematic analysis is crucial also to avoid tendinopathies (http://rua.ua.es/dspace/handle/10045/108922) and muscle injuries in such athletes (https://www.mdpi.com/2411-5142/6/3/75)

Response:

Thank you very much for the useful literature information.

We have added the following: (Page 1, Lines 33-38)

In fact, it has been suggested that elucidating individual muscle activity patterns [3] and joint stiffness [4] in complex movements such as running may lead to improvements in running behavior. Furthermore, it may provide useful information for the prevention of tendon disorders [5] that may have a negative impact on the performance of athletes and for rehabilitation approaches to muscle injuries [6]  .

Reviewer 3 Report

Thank you for the invitation to review this very interesting article. The topic is very interesting, since the evaluation of running strategies in patients with stroke is a poorly understood area. However, with respect to the paper, I have some concern, that I list below:

My first concern is about the subject description. Considering the fact that this is a case control study, the informations on the main and only subject should be deepen expecially with the type of stroke, the amount of physical activity and with an objective evaluation of the motor performance of the lower extremity of the subject. The presence of a single subject makes me believe that the significance of content and scientific soundness is low. 

Additionally, as you wrote in your limitations, even if gait analysis is a validated instrument to assess gait perfomance in stroke patients, it's use in clinical settings is very expensive and artificial respect to other innovative instruments like IMUs, given that i don't think that such a method should likely be further explored as a performance evaluation for para-athletes who are stroke patients this is why I think your results have low scientific soundness.   

Author Response

We are thankful for the essential questions and suggestions of the reviewers regarding the research contents. Our responses to the comments are as follows:

Comment:

My first concern is about the subject description. Considering the fact that this is a case control study, the informations on the main and only subject should be deepen expecially with the type of stroke, the amount of physical activity and with an objective evaluation of the motor performance of the lower extremity of the subject. The presence of a single subject makes me believe that the significance of content and scientific soundness is low.

Response:

Thank you very much for your important remarks.

Initially, we tried to consolidate the number of subjects as much as possible, but the degree of disability of stroke patients is very diverse, and the fact that they are para-athletes was the reason for the single case study. However, we believe that para-athletes, like athletes in general, require individualized measures of performance. Therefore, we proposed an evaluation method of joint movement coordination as a methodology in this study.

Comment:

Additionally, as you wrote in your limitations, even if gait analysis is a validated instrument to assess gait performance in stroke patients, it's use in clinical settings is very expensive and artificial respect to other innovative instruments like IMUs, given that I don't think that such a method should likely be further explored as a performance evaluation for para-athletes who are stroke patients this is why I think your results have low scientific soundness.

Response:

As you mentioned, we are not able to suggest a complete alternative method that can be proposed for 3D motion systems at this time. However, in the last few years, it has been shown that even inexpensive game console accessories available on the market can track joint trajectories with high color and depth resolution. In that report [reference1], the authors stated that the joint angles of hip and knee joints are more reliable than those of 3D motion capture systems, supporting the validity and reliability of gait analysis and potentially complementing clinical gait assessment. Therefore, we believe that our proposal also has future potential with respect to clinical evaluation.

Another objective of our study was to propose coherence analysis as an evaluation index for motor coordination of the lower limbs. We believe that such an index can be applied to slow movements such as walking, and it is known that the basic common structure of motor output (e.g., muscle activity) is very similar between walking and running [reference2], even though the speed is different. Therefore, we believe that our analysis method can be applied to basic research.

[reference1]

Latorre, J., Colomer, C., Alcañiz, M. et al. Gait analysis with the Kinect v2: normative study with healthy individuals and comprehensive study of its sensitivity, validity, and reliability in individuals with stroke. J NeuroEngineering Rehabil 16, 97 (2019).

[reference2]

Cappellini G, Ivanenko YP, Poppele RE, Lacquaniti F. Motor patterns in human walking and running. J Neurophysiol. 2006 Jun;95(6):3426-37. doi: 10.1152/jn.00081.2006. Epub 2006 Mar 22. PMID: 16554517.

We have added the following: (Page 6, Lines 229-233)

In fact, the 3D motion analysis system used in this experiment was very expensive, but recently, a very inexpensive gait analysis tool that evaluates joint angles has been pro-posed [34]. Therefore, evaluation outside the laboratory can be expected. Furthermore, we believe that our proposed method for analyzing the coordination of joint motion can be widely applied not only to the running evaluation of para-athletes and mild stroke pa-tients, but also to normal gait analysis.

Reviewer 4 Report

The subject of intersegmental coordination in human locomotion is interesting and current, as it has applications for individualized exercise prescription, as well as for interventions and evaluations that seek to reduce the risk of injuries during these activities. The study has an exploratory character and is a case study. Several parts of the text suggest that the text still needs some 'maturing', including a clear lack of references in some statements (which I cite in the minor points). I suggest indicating the pilot or case study character in the title of the paper.

The purpose of the study misleads the reader to believe that paraathletes (plural) will be analyzed when in fact it is one subject only. Adjust. The results in the abstract are difficult to interpret because stating that there are absence or peaks of coherence in the high-frequency band above 4Hz does not state whether this is normal or not. Here there is a need for a control subject. Clearly, the study has the characteristics of a pilot study, still far from more consistent scientific information. I suggest rejecting the article and suggesting a more traditional approach with a representative sample of these for athletes and a comparator control group.

the first sentence of the introduction presents a circular reasoning. Yes, the possibility to analyze running coordination, allows you to analyze motor control during running. The second sentence follows from the first, but again, it is poorly constructed, I suggest you indicate the studies that have found that determining aspects of motor control aided in improving running performance. The fourth sentence of the introduction is incomplete, I suggest adding the determination of mechanical work and mechanical efficiency (e.g., https://pubmed.ncbi.nlm.nih.gov/34197674/ and https://pubmed.ncbi.nlm.nih.gov/21613286/).

the third sentence on the second page also makes logical sense because intersegmental coordination is not a direct extension of intermuscular coordination assessed by EMG.

minor points, weight is in N, body mass is ni kg.

- consider using the residual analysis proposed by Winter (2009) instead of using a fixed cut-off frequency in the filter.

- 2nd paragraph of discussion - in e of (?)

Also, the rationale used in this paragraph is misleading, the co-contraction is mis-used here, I suggest strongly review all paragraph.

Author Response

We are thankful for the essential questions and suggestions of the reviewers regarding the research contents. Our responses to the comments are as follows:

Comment:

The subject of intersegmental coordination in human locomotion is interesting and current, as it has applications for individualized exercise prescription, as well as for interventions and evaluations that seek to reduce the risk of injuries during these activities. The study has an exploratory character and is a case study. Several parts of the text suggest that the text still needs some 'maturing', including a clear lack of references in some statements (which I cite in the minor points). I suggest indicating the pilot or case study character in the title of the paper.

Response:

In line with your suggestion, we have revised the title to make it clear that this study is a case study. Title :Lower Limb Kinematic Coordination during the Running Motion of Stroke Patient: A Single Case Study

Comment:

The purpose of the study misleads the reader to believe that paraathletes (plural) will be analyzed when in fact it is one subject only. Adjust. The results in the abstract are difficult to interpret because stating that there are absence or peaks of coherence in the high-frequency band above 4Hz does not state whether this is normal or not. Here there is a need for a control subject. Clearly, the study has the characteristics of a pilot study, still far from more consistent scientific information. I suggest rejecting the article and suggesting a more traditional approach with a representative sample of these for athletes and a comparator control group.

Response:

We have revised the way the unclear results are written in the abstract as follows This case report is a case study of one person with stroke and one healthy person, and when comparing the two, I considered the data of the healthy subject as "normal" and described the characteristics of the "stroke subject". We measured the running motion of several para-athletes, but because the symptoms in the stroke group were diverse and very difficult to statistically examine, we provided case reports rather than group comparisons.

Comment:

the first sentence of the introduction presents a circular reasoning. Yes, the possibility to analyze running coordination, allows you to analyze motor control during running. The second sentence follows from the first, but again, it is poorly constructed, I suggest you indicate the studies that have found that determining aspects of motor control aided in improving running performance.

Response:

Thank you for your suggestion.

We have added the following: (Page1, Lines 33-38)

In fact, it has been suggested that elucidating individual muscle activity patterns [3] and joint stiffness [4] in complex movements such as running may lead to improvements in running behavior. Furthermore, it may provide useful information for the prevention of tendon disorders [5] that may have a negative impact on the performance of athletes and for rehabilitation approaches to muscle injuries [6].

[3] Cappellini, G.; Ivanenko, Y.P.; Poppele, R.E.; Lacquaniti, F. Motor patterns in human walking and running. j. Neurophysiol. 2006, 95, J. Neurophysiol.

[4] Kuitunen S, Komi PV, Kyröläinen H. Knee and ankle joint stiffness in sprint running. Med Sci Sports Exerc. 2002 Jan;34(1):166-73. doi: 10.1097/00005768-200201000-00025. pmid: 11782663.

[5] Sirico. F.; Palermi. S.; Massa, B.; Corrado. B. Tendinopathies of the hip and pelvis in athletes: A narrative review. Journal of Human Sport and Exercise. 2020, 15, 3,748-762.

[6] Palermi, S.; Massa, B.; Vecchiato, M.; Mazza, F.; De Blasiis, P.; Romano, A.M.; Di Salvatore, M.G.; Della Valle, E.; Tarantino, D.; Ruosi, C.; Sirico, F. Indirect Structural Muscle Injuries of Lower Limb: Rehabilitation and Therapeutic Exercise. J. Funct. Morphol. Kinesiol. 2021, 6, 75.

Comment:

The fourth sentence of the introduction is incomplete, I suggest adding the determination of mechanical work and mechanical efficiency (e.g., https://pubmed.ncbi.nlm.nih.gov/34197674/ and https://pubmed.ncbi.nlm.nih.gov/21613286/).

Response:

Thank you for your suggestion.

In line with your suggestion, we have cited the following two references.

We have added the following: (Page1, Lines 38-42)

In the analysis of the running movements of competitive athletes, the collection of joint moments [7,8], joint angles [9], and ground reaction forces [10] through motion capture systems as well as the collection of lower limb muscle activity through electromyography [11] mechanics work [12] and mechanical efficiency [13] are commonly carried out in analyses.

[12] Peyré-Tartaruga LA, Dewolf AH, di Prampero PE, Fábrica G, Malatesta D, Minetti AE, Monte A, Pavei G, Silva-Pereyra V, Willems PA, Zamparo P. Mechanical work as a (key) determinant of energy cost in human locomotion: recent findings and future directions. Exp Physiol. 2021 Sep;106(9):1897-1908. doi: 10.1113/EP089313. Epub 2021 Jul 14. PMID: 34197674.

[13] Farris DJ, Sawicki GS. The mechanics and energetics of human walking and running: a joint level perspective. J R Soc Interface. 2012 Jan 7;9(66):110-8. doi: 10.1098/rsif.2011.0182. Epub 2011 May 25. PMID: 21613286; PMCID: PMC3223624.

Comment:

minor points, weight is in N, body mass is in kg.

Response:

We have corrected "weight" to "body mass" as you suggested.

Comment:

- consider using the residual analysis proposed by Winter (2009) instead of using a fixed cut-off frequency in the filter.

Response:

Thank you very much for the useful information. In this research, we conducted frequency analysis at the time of the preliminary experiment and determined the cutoff frequency based on the results, but we would like to work on the method you suggested.

Comment:

- 2nd paragraph of discussion - in e of (?)

Response:

Sorry for the confusing diagram. " The "e" is a symbol used to indicate a specific range in Figure 3.

Comment:

Also, the rationale used in this paragraph is misleading, the co-contraction is mis-used here, I suggest strongly review all paragraph.

Response:

We have edited "co-contraction" to "co-activation" in each paragraph of the discussion.

Round 2

Reviewer 4 Report

The authors replied all questions raised by me.

minor points

line 41 - mechanical work.

line 80 - left (double space before parenthesis).

lines 185 and 188 - Consider including references from literature to support this statement on healthy subjects, further than your subject analyzed.

- consider analyzing also mechanical results from strokes during walking, highlighting that your findings extend these results to running (e.g. PMID: 32694152).

Author Response

Responses to the comments of Review:4 (Round 2)

 We are grateful for further detailed essential questions and suggestions from the reviewers regarding our research. Our responses to the comments are as follows:

Comment:

line 41 - mechanical work.

Response:

Thank you for pointing this out. I have corrected it as instructed.

Comment:

line 80 - left (double space before parenthesis).

Response:

Thank you for pointing this out. I have corrected it as instructed.

Comment:

lines 185 and 188 - Consider including references from literature to support this statement on healthy subjects, further than your subject analyzed.

Response:

Thank you for your suggestion.

We have added the following: (Page 5, Lines185-189)

 It has been shown that in rhythmic movements, such as human walking, the frequencies of body acceleration are usually observed in the lower range [29]. Considering that higher frequencies are observed in pathological subjects, it is also interesting to note that a peak in coherence was observed in stroke patients, but not in healthy subjects.

[29] Garcia, F.D.;Da Cunha,M.J.;Schuch,C.P.;Schifino,G.P.;Balbinot,G.;Pagnussat,A.S. Movement smoothness in chronic post-stroke individuals walking in an outdoor environment—A cross-sectional study using IMU sensors. PLOS ONE. 2021, 16(4), e0250100. 

Comment:

- consider analyzing also mechanical results from strokes during walking, highlighting that your findings extend these results to running (e.g. PMID: 32694152).

Response:

Thank you for your suggestion.

We have added the following: (Page6, Lines 208-213)

Previous reports on mechanical energy during walking in stroke victims have reported the presence of neural connections between limbs, such as increased movement on the non-paralyzed side to stabilize the body against asymmetries in internal energy generation after stroke. Such a concept is probably also possible in running. Therefore, the results of this study in running may reflect the complex neural adaptations of the motor system seen after stroke [34].

[34] Balbinot,G.; Schuch,C.P.; Bianchi Oliveira ,H.;Peyré-Tartaruga,L.A. Mechanical and energetic determinants of impaired gait following stroke: segmental work and pendular energy transduction during treadmill walking. Biol Open. 2020, 21,9, bio051581.
